# A High-Sensitivity Dual-Axis Accelerometer with Two FP Cavities Assembled on Single Optical Fiber

**DOI:** 10.3390/s22249614

**Published:** 2022-12-08

**Authors:** Bolun Zheng, Hanjie Liu, Zhen Pan, Xi Chen, Lixiong Wang, Dian Fan

**Affiliations:** 1National Engineering Research Center of Fiber Optic Sensing Technology and Networks, Wuhan University of Technology, Wuhan 430070, China; 2School of Information Engineering, Wuhan University of Technology, Wuhan 430070, China

**Keywords:** accelerometer, Fabry–Pérot, optical fiber sensor, 2D, prism, high sensitivity

## Abstract

In this paper, a dual-axis Fabry–Pérot (FP) accelerometer assembled on single optical fiber is proposed. The sensor is equipped with a special beam-splitting prism to split the light into two perpendicular directions (the X- and Y-axes); the prism surface coated with semi-permeable film and the reflective sheet on the corresponding Be-Cu vibration-sensitive spring form two sets of FP cavities of different sizes. When the Be-Cu spring with a proof mass (PM) is subjected to the vibration signal, the cavity length of the corresponding FP cavity is changed and the interference signal returns to the collimator through the original path of the prism. After bandpass filtering and demodulation, the two cavity lengths are obtained, and the acceleration measurement in dual-axis directions is completed. The resonant frequency of the proposed dual-axis fiber optic accelerometer is around 280 Hz. The results of the spectral measurements show 3.93 μm/g (g = 9.8 m/s^2^: gravity constant) and 4.19 μm/g for the applied acceleration along the X- and Y-axes, respectively, and the cross-axis sensitivity is below 5.1%. Within the angle range of 180°, the maximum error of measured acceleration is less than 3.77%. The proposed fiber optic dual-axis FP accelerometer has high sensitivity and strong immunity to electromagnetic interference. The size of the sensor mainly depends on the size of the prism, which is easy to reduce and mass produce. Moreover, this FP construction method has high flexibility and development potential.

## 1. Introduction

With the popularity of the Internet of Everything and the increased demand for structural health monitoring, accelerometry is of great importance in the aerospace, automotive, and robotics industries, and has a wide range of applications in sensing and monitoring scenarios, such as bridges and railroads [1,2,3]. Accelerometers are generally divided into electrical accelerometers and optical accelerometers. Among them, fiber optic accelerometers have gained considerable attention and undergone rapid development due to their high sensitivity, corrosion resistance, anti-electromagnetic interference, and intrinsic safety [4]. Fiber optic accelerometers mainly contain two types: the wavelength type and the interferometric type. Fiber Bragg grating (FBG)-type accelerometers, as the classical type of wavelength type accelerometers, have been continuously improved since their first demonstration by Berkoff et al. [5,6,7]; however, their sensitivity is still lower when compared with interferometric accelerometers. Interferometric accelerometers exist in various types, such as Fabry–Pérot, Mach–Zehnder [8,9], Michelson [10,11], and Sagnac [12,13], among which fiber optic Fabry–Pérot sensors have the advantages of simple fabrication of FP cavity sensing head, good reliability, flexible fabrication, and many other advantages such as high sensitivity [14].

A typical fiber optic FP accelerometer structure is formed by connecting the spring and the PM together, and then pasting the reflective film on the PM to form an FP cavity with a fiber end face. When the external acceleration acts, the relative displacement between the PM and the fiber changes the length of the FP cavity due to the inertia effect, and the acceleration value of the measured object is obtained through the demodulation of the cavity length [15,16,17,18]. However, the number of FP cavities that can be constructed by a single fiber is extremely limited. In 2019, Xiaoying Liu et al. [19] proposed a compact fiber optic Fabry–Pérot (FP) sensor for the simultaneous measurement of acoustics and temperature, which consists of a silicon–glass–silica sandwich structure formed by anodic bonding, with a circular shape. The air FP cavity with circular holes and grooves utilizes the deformation of the silicon film for acoustic measurements, and the silicon FP cavity utilizes the temperature sensitivity of the silicon refractive index for temperature measurements. Using a multilayer reflective structure is a common method used to construct multiple FP cavities, so multiple parameters can be measured on the basis of a single fiber. However, multiple FP cavities are parallel to each other, which makes it difficult to expand to different directions in acceleration measurement. In 2021, Majid Taghavi et al. [20] used two optical fibers to construct separate FP cavities in the vertical direction to complete dual-axis acceleration measurements, which is the most common method of multi-dimensional acceleration measurements [21,22]. This is the most common method used to construct an FP cavity with an optical fiber end face alone in each axis direction, or directly combine multiple one-dimensional accelerometers into a whole to complete multi-dimensional measurement, but this method not only makes the sensor more vulnerable to the influence of fiber bending performance in multiple directions, but also makes the accelerometer require more fiber interfaces, which increase the complexity of the system.

This paper proposes a single fiber optic dual-axis FP accelerometer. The light source of the sensor enters the beam-splitting prism through the collimator, and the output surface of the beam-splitting prism is coated with semi-transmissive and semi-reflective film, which forms an FP cavity with the reflector at the spring. This method of forming an FP cavity provides a good platform for the design and installation of springs. The results of this paper show that the accelerometer we proposed can successfully accomplish the measurement of dual-axis acceleration; the resonant frequency of the accelerometer is about 280 Hz, and has a flat frequency response in the range of 0–120 Hz. Moreover, the X-axis and Y-axis sensitivities of the accelerometer reach 3.93 μm/g and 4.19 μm/g, respectively, at a vibration frequency of 100 Hz, and the error of the measured acceleration value does not exceed 3.77% within the measurement range of 180°. According to the results, this dual-axis accelerometer only needs one interface, has high sensitivity, and completes the two-dimensional direction measurement well. The structure of the sensor is simple and efficient, with the prism-splitting surface providing the role of vertical beam splitting. The output surface involved in the construction of the FP cavity while providing a platform for the construction of the spring. The FP cavity-forming method used in the accelerometer we proposed has great potential for mass production in the multi-dimensional sensing field.

## 2. Structure and Principle

Figure 1a shows a schematic diagram of the structure of the dual-axis FP accelerometer (sensor housing not included). The sensor consists of a collimator, a special optical prism, fan-shaped shims, Be-Cu springs, and PMs. The collimator is coated with a transmission-enhancing film. The incident surface of the prism is coated with a transmission-enhancing film, the spectroscope of the prism can divide the light with a wavelength range of 1500 nm to 1600 nm equally into two vertical directions, and the output side is coated with a 30% reflection and 70% transmission film. The fan-shaped shims are fixed onto the output side of the prism to provide a platform for the installation of the spring, and the thickness of the fan-shaped shim is used to control the cavity length of the FP cavity, so that the two FP cavities in the vertical direction have unequal cavity lengths. In Figure 1b, the collimator output surface is coated with a transmission-enhancing film, marked as R_1_; the two transmission-enhancing films found on the prism are marked as R_2_ and R_8_; the 50:50 splitting surface of the prism is marked as R_3_; and the two 30% reflection and 70% transmission films of the prism are marked as R_4_ and R_5_. The centers of the springs in two vertical directions are equipped with reflective sheets, which are marked as R_6_ and R_7_.

The prism’s semi-permeable film R_4_ and the corresponding directional spring reflective sheet R_6_ form FP_1_ with a cavity length of L_1_, and the prism’s semi-permeable film R_5_ and the corresponding directional spring reflective sheet R_7_ form FP_2_ with a cavity length of L_2_, where L_1_ is significantly smaller than L_2_. Since the prism is a cubic structure, the distances R_1_ to R_4_ and R_5_ are almost equal, which makes the optical path difference (OPD) of the interference signal between R_4_ and R_5_ close to 0, which can be ignored. There is Michelson interference between R_6_ and R_7_, and its arm difference is the difference between L_1_ and L_2_. The lights reflected from R4, R_5_, R_6_, and R_7_ interfere with each other and the total light intensity [20] returned to the collimator is
(1)$$Ι\left(\lambda \right)=I_{1}+I_{2}+I_{3}-2\sqrt{I_{1}I_{2}}\cos\left(\phi_{1}\right)-2\sqrt{I_{1}I_{3}}\cos\left(\phi_{2}\right)+2\sqrt{I_{2}I_{3}}\cos\left(\phi_{2}-\phi_{1}\right)$$

The intensity of I_1_, I_2_, and I_3_ refers to the light intensities of the three reflected lights, where I_1_ is the sum of the reflected light from R_4_ and R_5_, I_2_ is the reflected light from R_6_, and I_3_ is the reflected light from R_7,_ all of which are only related to the wavelength of the incident light λ; thus, they can be considered as a constant, approximately. From Equation (1), it can be seen that the reflection spectrum of the sensor is mainly a linear superposition of three groups of cosine functions, where the interference spectra of FP_1_ and FP_2_ correspond to the fourth and fifth terms of Equation (1). $\phi_{1}$ and $\phi_{2}$ are the phase shifts of FP_1_ and FP_2_, respectively, which can be expressed as
(2)$$\phi_{1}=\frac{4{\pi n}_{\mathrm{air}}L_{1}}{\lambda },\phi_{2}=\frac{4{\pi n}_{\mathrm{air}}L_{2}}{\lambda }$$
where n_air_ is the effective refractive index of air and is defaulted to 1. From Equation (2), it can be seen that $\phi_{1}$ and $\phi_{2}$ are positively correlated with cavity length, so in the process of making the sensor, the unequal size of the cavity lengths L_1_ and L_2_ can lead to unequal frequencies of the corresponding spectrum. The spectra of different frequencies correspond to different FP cavity lengths, so they can be separated by appropriate band-pass filtering to obtain the cavity lengths L_1_ and L_2_ for FP_1_ and FP_2_, respectively.

The cavity lengths of FP_1_ and FP_2_ are L_1_ and L_2_ when the sensor remains stationary. When excited by acceleration in the dual-axis plane, the springs with PMs produce axial movement which will cause the cavity length to change correspondingly; the cavity lengths of FP_1_ and FP_2_ change ${\Delta L}_{1}\left(t\right)$ and ${\Delta L}_{2}\left(t\right)$. Existing literature studies have shown that there is a linear relationship between acceleration and the change in cavity length of the FP cavity of optical fiber [23]. The acceleration value is linearly related to the cavity length of the fiber optic FP cavity as follows:(3)$${\Delta L}_{1}\left(t\right)=S_{1}a_{1}\left(t\right),{\Delta L}_{2}\left(t\right)=S_{2}a_{2}\left(t\right)$$
where $S_{1}$ and $S_{2}$ are the sensitivities of the FP_1_ and FP_2_ spring, that is, the amount of change in cavity length after receiving an acceleration of 1 g. a_1_(t) and a_2_(t) correspond to the acceleration of the *X*-axis and *Y*-axis, respectively, and the relationship between a_1_(t) and a_2_(t) is shown in Figure 2. The following sections will take θ, which is the angle between the *X*-axis of the sensor and the ground, as the angle reference.

The directions of a_1_(t) and a_2_(t) are perpendicular to each other, and thus the total acceleration values a(t) can be expressed as
(4)$$a\left(t\right)=\sqrt{a_{1}^{2}\left(t\right)+a_{2}^{2}\left(t\right)}$$

It is worth noting that in the later experiment, the direction of the total acceleration a(t) is perpendicular to the ground. Therefore, the following equation can be deduced:(5)$$a_{1}\left(t\right)=a\left(t\right)\sin\theta ,a_{2}\left(t\right)=a\left(t\right)\cos\theta$$

The prism is a 10 mm × 10 mm × 10 mm sized cube, the incident surfaces R_2_ and R_8_ are coated with a transmission-enhancing film to avoid crosstalk on the interference signal, and the beam-splitting surface R_3_ uses 50:50 vertical beam splitting to divide the incident light equally into two vertical directions. In order to make the reflected light intensity of different reflective surfaces approximately equal, so that the interference spectrum has good contrast, the two output surfaces R_4_ and R_5_ are coated with a 30% reflection and 70% transmission film. The Be-Cu springs are square with a thickness of 0.05 mm and side length of 10 mm, which matches the size of the prism. The spring is designed symmetrically to increase its stability against different angles. A 0.1 g PM is glued to one side of the spring to increase its sensitivity, and a reflective sheet is glued to the other side of the spring to form the FP cavity with the prism’s corresponding output surface; the two directional springs are made in the same way.

The sensor is fabricated as follows: First, using the sensor housing as a medium, the relative positions of the collimator and prism are fixed to ensure that the reflected light from the prisms R_4_ and R_5_ can return to the collimator. The four equally thick fan-shaped shims are glued to the four corners of the prism’s output side, which not only help to control the length of the FP cavity, but also help to maintain the parallelism between the prism’s output side and the reflective sheet of the spring, so that the desired FP and interference spectrum can be obtained. To ensure the interference spectra of the two FP cavities can be separated, it is important that the thickness of the shims in the two vertical directions is different, which means that L_1_, L_2_, and L_2_ − L_1_ are not equal. Specifically, the thickness of the shims is about 650 for FP_1_ and 1750 for FP_2_. Finally, the completed springs are glued on the fan-shaped shims, and the internal structure of the sensor is shown in Figure 1c. After packaging, the appearance of the sensor is as shown in Figure 1d.

## 3. Experiment and Results

Figure 3a shows a diagram of the sensor test experiment system, and the structure in the red wireframe is shown in Figure 3b. The vibration experiment uses a Danish B&K vibration test platform with a frequency excitation range of 5–1000 kHz. The reference sensor uses a piezoelectric accelerometer from B&K with a resonant frequency of 16 kHz, a sensitivity of 10 pC/ms^−2^, and a measurement range of 0.1–4800 Hz. The broadband light generated by the broadband light source (BBS) reaches the optical sensor through the circulator, and the optical sensor is fixed on the rotating platform, which has an adjustment range of 90°. The frequency and amplitude of the excitation signal are controlled by the PC and power amplifier, respectively, so that the vibration exciter produces the expected vibration excitation. Meanwhile, the vibration signal is recorded by the electrical accelerometer on the vibration platform. The optical accelerometer, that is also excited, reflects the signal light back to the circulator and reaches the demodulator, which sends the spectral information of 1530 nm–1560 nm to the PC for storage and subsequent demodulation at an acquisition frequency of 2000 Hz.

The reflection spectra of the optical fiber FP sensor is shown in Figure 4a, which is the result of mixing multiple groups of interference spectra. Figure 4b shows the result of the FFT transformation of the reflection spectra, where there are three distinct frequency components corresponding to peaks 1, 2, and 3, respectively, and half of the OPDs corresponding to the three peaks are demodulated with values of 613, 1152, and 1765, respectively. Combined with the structure of the sensor, we can draw the conclusion that peak 1 corresponds to FP_1_ and its cavity length is L_1_, and peak 3 corresponds to FP_2_ and its cavity length is L_2_. The incident light is divided into two beams by the beam-splitter and reflected back by the corresponding reflective surfaces R6 and R7, forming a Michelson interference. According to Figure 1b, peak 2 corresponds to the Michelson interference phenomenon formed between R_6_ and R_7_, with the value of the arm length difference L_3_ equal to the difference between the cavity length L_2_ and L_1_. Figure 4c,d show the interference spectra corresponding to FP_1_ and FP_2_ alone after bandpass filtering. Due to the influence of the filter performance, the initial position amplitude of the spectrum is attenuated. However, this has little impact on the subsequent demodulation process, because the demodulation principle in this paper is based on the phase of the spectrum rather than the amplitude.

### 3.1. Frequency Response

In order to determine the resonant frequency and operating frequency range of the fiber optic accelerometer, the response of the sensor at different vibration frequencies needs to be measured. Adjust θ to 90° by rotating the platform, apply the vibration signal to FP_1_, adjust the vibration amplitude of the exciter to keep it constant and scan the frequency from 20 Hz to 480 Hz in steps of 20 Hz, record the response of FP_1_, and ignore the response of FP_2_. Then, adjust θ to 180° and repeat the same operation to obtain the response of FP_2_.

The frequency response of the optical sensor is shown in Figure 5, and the experimental results show that the resonance frequencies of the optical sensor were obtained as about 280 Hz for both the X- and Y-axes. These values in the two directions are slightly different. This difference can be attributed to a slight weight difference between the PMs and a machining error of springs. Considering the results of frequency response in both directions, we can see that the sensor has a fairly flat frequency response when the frequency is lower than 100 Hz; thus, the flat region 0~100 Hz is taken as the operating frequency range.

### 3.2. Sensitivity Calibration

To determine the correspondence between the cavity length change and the acceleration value in two directions of the dual-axis FP accelerometer, the sensitivity of the springs on both axes was calibrated separately by adjusting the angle of the sensor. In the same way as described above, adjust θ to 90° to calibrate FP_1_ and adjust θ to 180° to calibrate FP_2_. The PC controls the vibration exciter, applies a sinusoidal vibration signal of a specified frequency to the sensor, adjusts different vibration amplitudes through the power amplifier knob, records and demodulates the FP cavity length change in the corresponding direction, compares the amplitude with the results recorded by the electrical accelerometer, and then obtains the single-sided sensitivity of the optical fiber accelerometer at the corresponding frequency. Figure 6 shows the linear response of the sensor to the *X*-axis and *Y*-axis at 20 Hz and 100 Hz. It can be seen that the linearity of the sensor’s response to the vibration signal is very good, and is higher than 99.9%. The corresponding results of amplitude–acceleration at 100 Hz are shown in Figure 6b, and the sensitivity of FP_1_ and FP_2_ are 3.93 μm/g and 4.19 μm/g, respectively. Compared to the value 3.22 nm/g reported in [20], the sensor shows a much higher sensitivity. The sensitivity of the two axes is slightly different, which may be due to the error in the spring processing.

### 3.3. Cross-Axis Sensitivity

For accelerometers, especially multi-dimensional accelerometers, it is necessary to consider not only the response of the sensor in the sensitive direction, but also the effect of the vibration component in the non-sensitive direction on the measurement in the sensitive direction. Adjust θ to 90° and 180°, respectively, and record and demodulate the sizes of the optical sensor’s FP cavities L_1_ and L_2_ in both directions. The cross-sensitivity of the sensor can be obtained by comparing the peak amplitude of two FP cavity length changes.

Figure 6 shows the response of the sensor to the vibration signal in the sensitive direction and the non-sensitive direction at 100 Hz vibration frequency. Figure 7a shows the change in cavity length of FP_1_ and FP_2_ when the *X*-axis of the sensor is vibrated. Average the wave crest and wave trough, respectively, calculate the difference between them to obtain the maximum displacement of FP cavity, and compare the maximum displacement of FP_1_ and FP_2_ to obtain the cross-sensitivity. The results show that the cross-axis sensitivity of the sensor is 4.9%. Figure 7b shows the change in cavity length of FP_1_ and FP_2_ when the *Y*-axis of the sensor is vibrated, and the results show that the cross-axis sensitivity of the sensor is 5.1%.

Considering the structure of the spring, the main reason why the cross-sensitivity of the sensor is slightly high is that when the PM and the spring are bonded as a whole, their center of gravity is on the PM instead of the spring. This structural feature makes it difficult for the spring to keep stable when it is subjected to vibration in the cross-direction, thus producing unexpected shaking, which leads to the increase in cross-sensitivity.

### 3.4. Multi-Angle Measurement

When adjusting the PC and power amplifier, keep the vibration frequency at 100 Hz, refer to the value of the electrical accelerometer, and keep the maximum span of the sine wave crest and wave trough at 0.697 g. Take the angle between the X-axis of the sensor and the ground as a reference, rotate the optical accelerometer from 45° to 225° in sequence by adjusting the rotating platform in steps of 5°, and record spectral signals in turn. As the rotation range of the rotating platform is 90°, the sensor needs to be re-fixed when rotating from 45° to 135° to test the 135° to 225° range.

It can be seen from Figure 6c,d that the sensitivities of FP_1_ and FP_2_ are known to be 3.93 μm/g and 4.19 μm/g, respectively. Based on the variation in the cavity lengths of the two FP cavities and their respective sensitivities, the accelerations of a_1_(t) and a_2_(t) at different angles are calculated and the total acceleration a(t) is finally obtained based on Equation (4). The span of the acceleration amplitude is calculated according to the difference between the wave crest and wave trough of the measured acceleration. The acceleration of the X-axis, Y-axis and the total acceleration at different measuring angles are shown in Figure 8. It should be noted that the angle range from 45° to 225° are measured data, and the other half are mirror data for better observation. At 90°, the acceleration measured by FP_1_ is close to the maximum value, and the acceleration measured by FP_2_ is close to 0, which is reversed at 180°. The acceleration amplitudes measured by FP_1_ and FP_2_ are approximately equal at around 45°, 135°, and 225°. The experimental results are consistent with Equation (5) and show that the dual-axis FP accelerometer responds effectively in the angle range of 45° to 225°, and the maximum error of the measured acceleration value does not exceed 3.77% when comparing its measured acceleration with that of the electrical accelerometer. When the angle is between 130° and 135°, FP_1_ and FP_2_ showed a poor continuity of response due to the disassembly and re-fixing of the sensor, as they were rotated to that angle.

## 4. Conclusions

In this paper, a dual-axis FP accelerometer is proposed and tested. The sensor can complete two-dimensional acceleration measurement. The structure of the sensor and its measurement principle are presented and analyzed. The sensor divides the light signal vertically by a prism, and uses the prism light output surfaces and the reflectors on the spring to form two FP cavities. The displacement of the springs are measured by two FP interferometers, respectively. A number of performance indicators of the sensor are tested using a vibration measurement platform. The resonant frequency of the proposed dual-axis fiber optic accelerometer is around 280 Hz, and the sensitivities in the sensitive direction at 100 Hz are 3.93 μm/g and 4.19 μm/g, respectively. The cross-axis sensitivity of this sensor is below 5.1%, and the maximum error of the measured acceleration value does not exceed 3.77% within the measurement range of 180°. The sensor was characterized by its high sensitivity, and only a single optical fiber is required in a two-dimensional measurement environment. In addition, the size of the sensor is mainly limited by the size of the prism, which is easy to reduce. In this paper, the two FP cavities are constructed using a simple and efficient method with high malleability, which is a potential method in the field of multi-dimensional FP sensing.

## Figures and Tables

**Figure 1 sensors-22-09614-f001:**
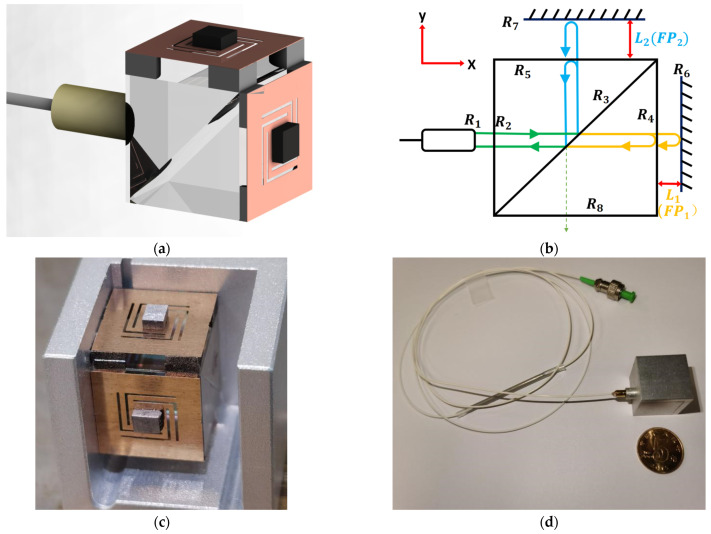
(**a**) Schematic diagram of the sensor; (**b**) interference mechanism of the sensor; (**c**) internal structure diagram of sensor; (**d**) packaged FP sensor. Schemes follow another format.

**Figure 2 sensors-22-09614-f002:**
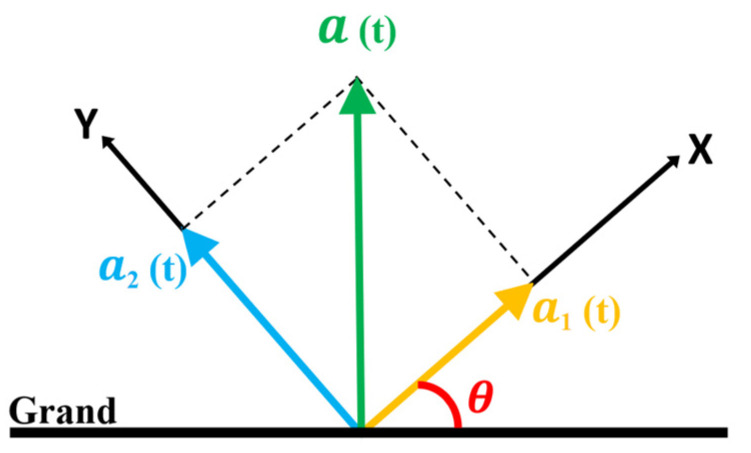
Relationship between measured accelerations.

**Figure 3 sensors-22-09614-f003:**
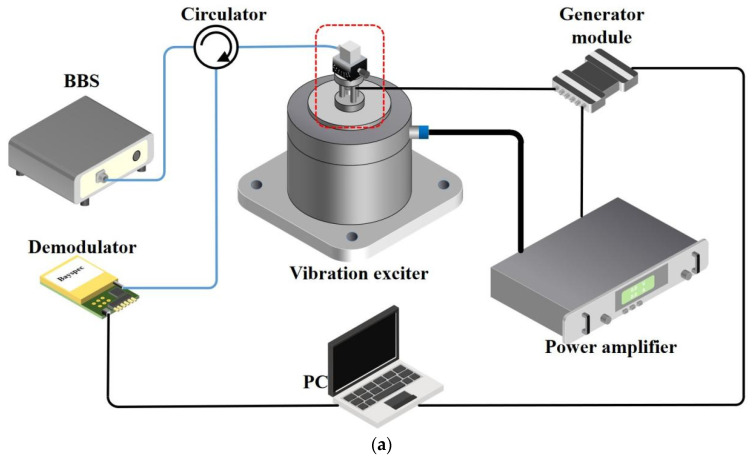
(**a**) Schematic diagram of the sensing system configuration; (**b**) partial enlarged view above the exciter.

**Figure 4 sensors-22-09614-f004:**
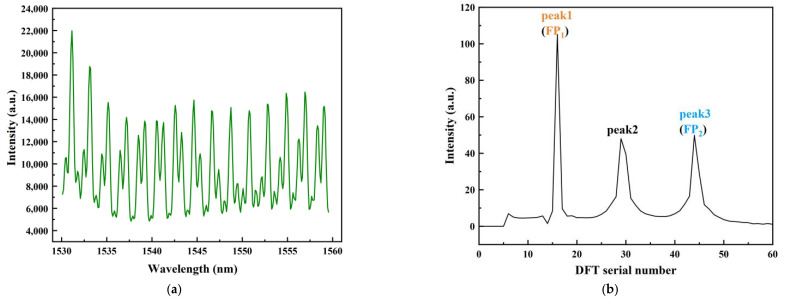
(**a**) Reflection spectra of the sensor; (**b**) spatial frequency spectra of the reflection light; (**c**) independent interference spectra of FP_1_ after filter; (**d**) independent interference spectra of FP_2_ after filter.

**Figure 5 sensors-22-09614-f005:**
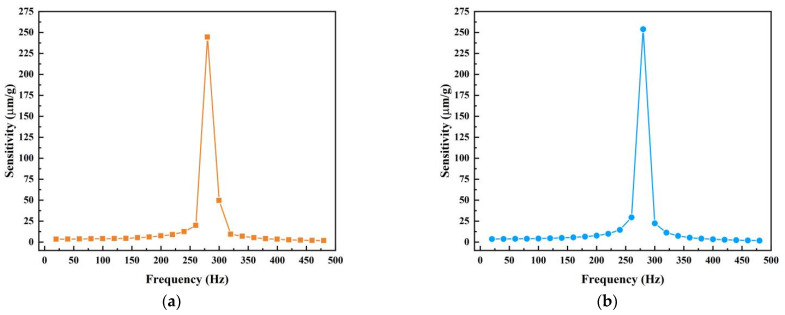
(**a**) The frequency response of FP_1_; (**b**) the frequency response of FP_2_.

**Figure 6 sensors-22-09614-f006:**
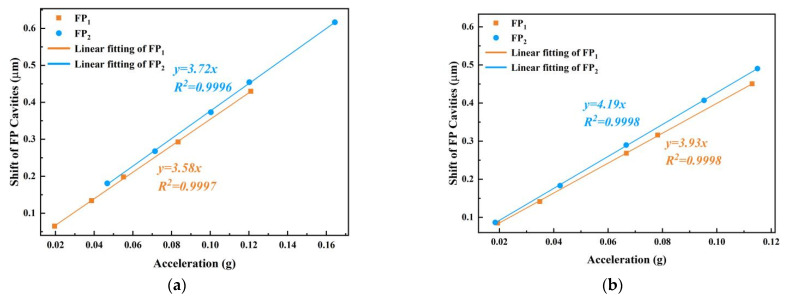
(**a**) Sensitivity along the X-axis and the Y-axis at 20 Hz; (**b**) sensitivity along the X-axis and the Y-axis at 100 Hz.

**Figure 7 sensors-22-09614-f007:**
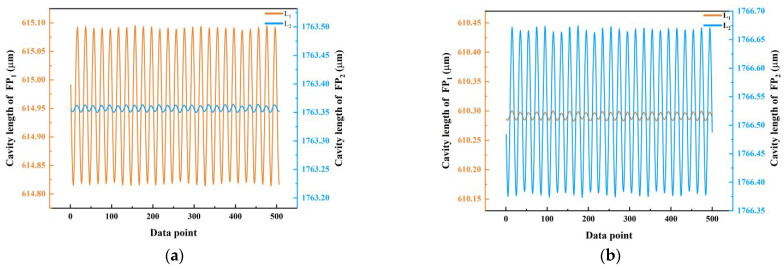
(**a**) The cavity length of FP_1_ and FP_2_ when θ=90°; (**b**) the cavity length of FP_1_ and FP_2_ when θ=180°.

**Figure 8 sensors-22-09614-f008:**
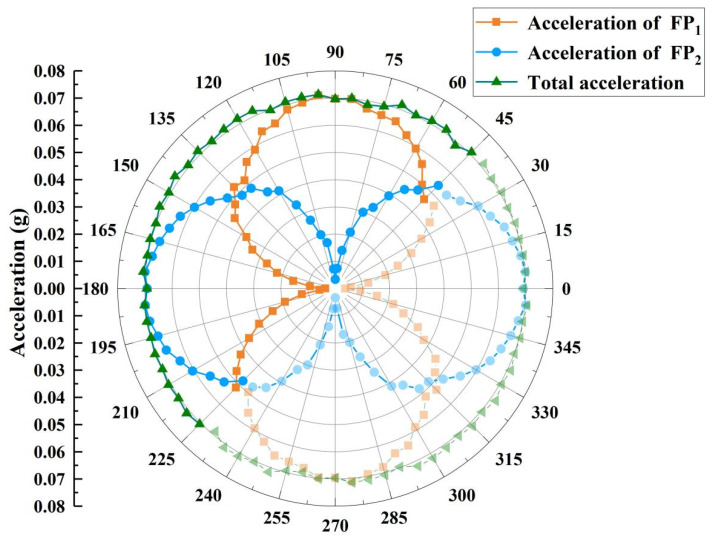
Response to the same acceleration in the range of 45°–225°.

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
