# Peer review of "A High-Sensitivity Dual-Axis Accelerometer with Two FP Cavities Assembled on Single Optical Fiber"

_sensors, 2022, doi:10.3390/s22249614_

Round 1

Reviewer 1 Report

The work by B. Zheng et al. reports on an optical fiber based dual-axis accelerometer, which exploits the realization of two Fabry-Perot cavities realized on a beam-splitter integrated on the optical fiber tip.

The work is well written and it could be of interest for the readers working in the field of optical fiber opto-mechanical sensors. However, before considering this work for a publication in Sensors, I would invite the authors to better highlight the novelty content in the introduction section, also taking into account recent works published in the same field, for example that by F.A. Bruno et al. (DOI: 10.1109/JLT.2019.2961766), which seems to use a similar spring-based FP cavity.

The impact of the results can be better highlighted, by explicitly comparing the obtained performances (for example the sensitivity and the bandwidth) with those already reported in the state of the art.

In addition, I invite the authors to enrich the technical content of the whole paper by also addressing the following points  :

- In the abstract the authors state that “The size of the sensor mainly depends on the size of the prism, which is easy to reduce and mass production. To what size scale can the prism be reduced, by taking into account the performances degradation and the fabrication processes? Can be it compatible with extremely compact and fully integrated solutions based on direct fabrication methods? (cfr. DOI: 10.1109/OMN.2015.7288865, DOI: 10.1039/D0NR00040J, DOI: 10.1016/j.snb.2020.128981, DOI: 10.1016/j.rio.2020.100051, DOI: 10.1364/OL.393900, just to name a few)

- How the spring is designed and realized? How do the spring characteristics influence the performances, for example in terms of central frequency and bandwidth of the resonant peak? And what about the proof mass?

- How the reflectivity spectra shown in Figure 4 have been evaluated? I would have expected values assumed in the range 0-1 (reflected light over incident light). In this regard, how the amplitude and thus the SNR influence the signal demodulation?

- The authors opted for the realization of low-finesse FPs, which make necessary the use of a filter for the signal demodulation. How the use of the filter ‘slows down’ the demodulation process, and how this aspect influence the effective bandwidth? Moreover, High-finesse solution FP characterized by sharp peaks located at specific wavelengths could in principle help to avoid the use of filters by exploiting the intrinsic wavelength selectivity (cfr. for example DOI: 10.1016/j.sna.2019.111795, DOI: 10.1038/s41566-017-0027-x ). The authors are invited to comment about this aspect.

- Which are the characteristics of the accelerometer used as a reference?

- The authors are invited to comment their results also in perspective of the possible application fields. In this regard, is the flat-band expected to be extended also below 20Hz (range required for seismic application)? And how far the band can be extended toward higher frequencies, taking into account the different aspects involved such as spring, mass, size, signal post-processing?

Author Response

Response to the Reviewers’ Comments

*********************************************************************

Reviewer #1: The work by B. Zheng et al. reports on an optical fiber based dual-axis accelerometer, which exploits the realization of two Fabry-Perot cavities realized on a beam-splitter integrated on the optical fiber tip. The work is well written and it could be of interest for the readers working in the field of optical fiber opto-mechanical sensors.

Comment #1.  However, before considering this work for a publication in Sensors, I would invite the authors to better highlight the novelty content in the introduction section, also taking into account recent works published in the same field, for example that by F.A. Bruno et al. (DOI: 10.1109/JLT.2019.2961766), which seems to use a similar spring-based FP cavity.

Response: Thank you very much for your careful review. We have added a description of sensor characteristics in the introduction to highlight innovation. And the article you provided really inspired the spring design of this paper, this reference is based on the micro-opto-mechanical (MOM) structure, which is different from this paper. We are sorry to have missed it before. Now, it has been added to the references.(Line 73-74, 81-82, 372-373)

Comment #2. The impact of the results can be better highlighted, by explicitly comparing the obtained performances (for example the sensitivity and the bandwidth) with those already reported in the state of the art.

Response: Thank you very much for your careful suggestion. We have added a comparison with another dual-axis accelerometer in the sensitivity calibration section. (Line 242-243.)

Comment #3In the abstract the authors state that “The size of the sensor mainly depends on the size of the prism, which is easy to reduce and mass production. To what size scale can the prism be reduced, by taking into account the performances degradation and the fabrication processes? Can be it compatible with extremely compact and fully integrated solutions based on direct fabrication methods? (cfr. DOI:10.1109/OMN.2015.7288865, DOI: 10.1039/D0NR00040J, DOI: 10.1016/j.snb.2020.128981, DOI: 10.1016/j.rio.2020.100051, DOI: 10.1364/OL.393900, just to name a few)

Response: Thank you very much for your careful review. According to the investigation on the spectroscopic prism on the market, its size can reach the millimeter level. The sensor in this paper can be assembled manually because the prism size is 10mm. Mechanized installation is necessary if smaller dimensions are to be achieved. Of course, with the reduction of prism size, the spring also needs to be redesigned. At present, the prism and spring module are designed independently in the sensor design, which have no influence on each other and are modularized. After the completion of production, they will be assembled together. To reach a fully integrated solutions, Springs and spacers may require a more compact and integrated design. However, the only requirement for the spring is that it can reflect light, which gives the spring great freedom in design. This paper focuses on verifying the feasibility of the principle and scheme, and will further try the design, fabrication and research of compact sensors to obtain ultra-small two-dimensional accelerometers.

Comment #4. How the spring is designed and realized? How do the spring characteristics influence the performances, for example in terms of central frequency and bandwidth of the resonant peak? And what about the proof mass?

Response: Thank you very much for your careful review. The rectangular cantilever beam was considered when designing the spring, but because of the requirement of multi angle response, the symmetrical spring design was finally selected. Factors such as the material, thickness, shape of the beam, weight of PM and even the installation method of PM will have an impact on the resonant frequency, bandwidth, sensitivity and other factors of the spring. Therefore, there are still many things that need to be optimized in the design of springs in this paper. Research on springs is indeed important, but this is not the focus of this article. Of course, we took your suggestions into account and added some discussion. (Line 159-160.)

Comment #5. How the reflectivity spectra shown in Figure 4 have been evaluated? I would have expected values assumed in the range 0-1 (reflected light over incident light). In this regard, how the amplitude and thus the SNR influence the signal demodulation?

Response: Thank you very much for your careful review. We are sorry there is an error in the Figure 4(a). The right Figure has been used to replace the error Figure in order to consistent with Figure 4 (b) (c) and (d). When we edited the paper, we got the wrong figure because the previous figure (a) was very similar with the Reflection spectra of the sensor beside the ordinate. The previous figure was preprocessed which is the result of Z-score normalization. The processed data conform to a standard normal distribution, i.e. the mean is 0 and the standard deviation is 1, but the magnitude is not limited to 0-1.It has now been replaced with the correct figure which is the data collected directly though the demodulator and it's the source of the data for the figure4 (b) (c) and (d). The ordinate units are unified as a.u. (arbitrary unit). The reason for using this unit is that different instruments have different representation methods, like light energy or photon counting, and there is no comparison between the values measured by different instruments, so the absolute value is generally not used, but expressed as a relative value. (figure 4(a))

As the question of the SNR influence the signal demodulation, when designing the parameters of the prism, we took into account its influence on the spectral SNR. Therefore, we adjust the reflectivity of the prism to make the reflected light from different reflective surfaces have a nearly equal light intensity (the reflectivity of the spring defaults to 1), thereby improving the fringe contrast of the interference spectrum and improving the SNR to avoid its influence on demodulation. The SNR of the signal is generally more than 5dB. With these methods, the quality of the signal is good, as shown in Figure 4(a). In response to this, we have added a more detailed description of the prism parameters section of the paper. (Line 155-157.)

Comment #6. The authors opted for the realization of low-finesse FPs, which make necessary the use of a filter for the signal demodulation. How the use of the filter ‘slows down’ the demodulation process, and how this aspect influence the effective bandwidth? Moreover, High-finesse solution FP characterized by sharp peaks located at specific wavelengths could in principle help to avoid the use of filters by exploiting the intrinsic wavelength selectivity (cfr. for example DOI: 10.1016/j.sna.2019.111795, DOI: 10.1038/s41566-017-0027-x ). The authors are invited to comment about this aspect.

Response: Thank you very much for your careful suggestion. The speed of demodulation does exactly need to be optimized in the future. We appreciate your suggestions on demodulation and this is a great inspiration for us. According to the paper you provided, we have noticed that using a single-wavelength laser as a light source is indeed a potential approach, but since the sensor in this paper contains more FP cavities. Because multiplexing of 2 FP cavities is involved, filter is difficultly to be avoided if you want to obtain high-precision results. We will later improve the demodulation speed through fast demodulation methods such as FPGA technology.

Comment #7. Which are the characteristics of the accelerometer used as a reference?

Response: Thank you very much for your careful review. The performance reference indexes of accelerometer include: working frequency, sensitivity, measurement range, etc. In this paper, the reference sensor uses a piezoelectric accelerometer from B&K with a resonant frequency of 16 kHz, a sensitivity of 10 pC/ms^-2, and a measurement range of 0.1-4800 Hz. And we have added a description of the reference sensor to the paper. (Line 178-180.)

Comment #8. The authors are invited to comment their results also in perspective of the possible application fields. In this regard, is the flat-band expected to be extended also below 20Hz (range required for seismic application)? And how far the band can be extended toward higher frequencies, taking into account the different aspects involved such as spring, mass, size, signal post-processing?

Response: Thank you very much for your careful review. The flat-band has the potential to expand below 20Hz, because we measured 10Hz at the lowest in the experiment, but we did not put the results in the paper, because the demodulated cavity length curve is sinusoidal, but its stability has declined. As you can see, the optical element of this sensor has no strong correlation with the moving element directly, which means that the performance of the spring is decisive when measuring vibration. By adjusting the spring material, thickness, shape, PM quality and other factors, we can change the resonant frequency of the spring and whether it is a spring with high resonant frequency or a spring with low resonant frequency, as long as it contains a reflector, it can theoretically be installed on the prism to work. However, in practical applications, there are still many things to consider, such as the sampling frequency of the instrument, demodulation methods, etc., so we are sorry that we cannot clearly indicate how high the passband can be expanded at this time. 

Reviewer 2 Report

1) Michelson interference; analysis

2) Equation 1; explain

3) Equation 2; analysis

4) ΔL(t)=S2a2(t); analysis

5) The sensitivity in the sensitive direction at 100Hz is 3.93 μm/g; explain

Author Response

Response to the Reviewers’ Comments

*********************************************************************

Reviewer #2

Comment #1. Michelson interference; analysis

Response: Thank you very much for your careful review. We have added a description of the formation of Michelson interference to the paper. (Line 198-200)

Comment #2. Equation 1; explain

Response: Thank you very much for your careful suggestion. We have added an explanation of the equation after equation (1) in the paper. (Line 117-122)

Comment #3Equation 2; analysis

Response: Thank you very much for your careful review. We have added an analysis of the equation after equation (2) in the paper and highlighted the reason why the different FP cavity lengths of the sensor are designed to be unequal according to equation 2. (Line 124-127)

Comment #4. ΔL(t)=S2a2(t); analysis

Response: Thank you very much for your careful review. We added literature citations to show that the magnitude of the acceleration is proportional to the change in the FP cavity. (Line 134-136)

Comment #5. The sensitivity in the sensitive direction at 100Hz is 3.93μm/g; explain

Response: Thank you very much for your careful review. This sensitivity value is based on the linear fit in Figure 6(b) and represents the displacement distance of the cavity length of the sensor after receiving a unit acceleration value of 1 g. 

Round 2

Reviewer 1 Report

I appreciate the effort of the authors to improve the paper on the basis of the comments raised.

All my doubts have been clearified, therefore I feel confident in suggesting this work for a publication in Sensors.